# Safety, Efficacy and Long-Term Outcomes of Patients Treated with the Occlutech Paravalvular Leak Device for Significant Paravalvular Regurgitation

**DOI:** 10.3390/jcm11071978

**Published:** 2022-04-01

**Authors:** Eustaquio Maria Onorato, Francesco Alamanni, Manuela Muratori, Grzegorz Smolka, Wojtek Wojakowski, Piotr Pysz, Aleksejus Zorinas, Diana Zakarkaite, Hélène Eltchaninoff, Pierre-Yves Litzer, François Godart, Patrick Calvert, Christos Christou, Abdurashid Mussayev, Bindo Missiroli, Igor Buzaev, Salvatore Curello, Tullio Tesorio, Antonio Luca Bartorelli

**Affiliations:** 1Centro Cardiologico Monzino, IRCCS, 20138 Milan, Italy; francesco.alamanni@ccfm.it (F.A.); manuela.muratori@gmail.com (M.M.); antonio.bartorelli@ccfm.it (A.L.B.); 23rd Division of Cardiology, Medical University of Silesia, 40-055 Katowice, Poland; gsmolka@me.com (G.S.); wwojakowski@sum.edu.pl (W.W.); piotr.pysz@gmail.com (P.P.); 3Department of Cardiovascular Medicine, Vilnius University, 01513 Vilnius, Lithuania; azorinas@gmail.com (A.Z.); diana.zakarkaite@santa.lt (D.Z.); 4Department of Cardiology and Cardiovascular Surgery, Hospital Charles Nicolle, 76000 Rouen, France; helene.eltchaninoff@chu.rouen.fr (H.E.); pierre-yves.litzler@chu-rouen.fr (P.-Y.L.); 5Department of Pediatric Cardiology and Congenital Heart Disease, University of Lille, 59000 Lille, France; francois.godart@chru-lille.fr; 6Department of Cardiology, Papworth Hospital NHS Foundation Trust, Cambridge CB2 0AY, UK; patrick.calvert1@nhs.net; 7American Heart Institute, Nicosia 1311, Cyprus; christos@amc.com.cy; 8National Research Cardiac Surgery Center, Astana 020000, Kazakhstan; abdurashid.mussayev@gmail.com; 9Gemelli Molise di Campobasso-Fondazione Policlinico Universitario Agostino Gemelli, IRCCS, 86100 Campobasso, Italy; bindo.missiroli@gemellimolise.it; 10Cardiovascular Department, Bashkir State Medical University, 450008 Ufa, Russia; igor@buzaev.com; 11USVD Emodinamica, Spedali Civili di Brescia, 25123 Brescia, Italy; scurello@libero.it; 12Department of Invasive Cardiology, Clinica Montevergine, 83013 Mercogliano, Italy; tulliotesorio@gmail.com; 13Department of Biomedical and Clinical Sciences “Luigi Sacco”, University of Milan, 20122 Milan, Italy

**Keywords:** paravalvular leak, paravalvular regurgitation, aortic valve, mitral valve, transcatheter closure, long-term outcomes, hemolysis, device

## Abstract

Between December 2014 and March 2021, 144 patients with aortic (Ao) or mitral (Mi) paravalvular leaks (PVLs) were enrolled at 21 sites in 10 countries. Safety data were available for 137 patients, who were included in the safety analysis fraction (SAF), 93 patients with Mi PVLs and 44 patients with Ao PVLs. The full analysis set (FAS) comprised 112 patients with available stratum (aortic/mitral leak) as well as baseline (BL), 180-day or later assessments (2 years). Procedural success (implantation of the device with a proper closure of the PVL, defined as reduction in paravalvular regurgitation of ≥one grade as assessed by echocardiography post implantation) was achieved in 91.3% of FAS patients with Mi PVLs and in 90.0% of those with Ao PVLs. The proportion of patients suffering from significant or severe heart failure (HF), classified as New York Heart Association (NYHA) class III/IV, decreased from 80% at baseline to 14.1% at 2-year follow-up (FAS). The proportion of FAS patients needing hemolysis-related blood transfusion decreased from 35.5% to 3.8% and from 8.1% to 0% in Mi patients and Ao patients, respectively. In total, 35 serious adverse events (SAEs) were reported in 27 patients (19.7%) of the SAF population. The SAEs considered possibly or probably related to the device included device embolization (three patients), residual leak (two patients) and vascular complication (one patient). During follow-up, 12/137 (8.8%) patients died, but none of the deaths was considered to be device-related. Patients implanted with the Occlutech Paravalvular Leak Device (PLD) showed long-lasting improvements in clinical parameters, including NYHA class and a reduced dependency on hemolysis-related blood transfusions.

## 1. Introduction

PVLs are the result of an incomplete seal between the sewing ring of the prosthetic valve and native annulus and occur more frequently after surgical mitral valve replacement (MVR) (up to 17%) compared to after aortic valve replacement (AVR) (up to 10%) [1,2,3,4,5,6]. Severe PVLs (1–5% of patients) are associated with dyspnea, left ventricular enlargement, heart failure (HF) and symptomatic hemolytic anemia (HA). In most centers, transcatheter PVL closure in symptomatic patients is taken into consideration in those at high risk for surgery [7,8,9,10,11,12,13,14,15,16]. Since 2014, catheter-based PVL closure with the PLD (Occlutech, GmbH, Jena, Germany) has been accomplished with the creation of an international expert group, on-going elaboration of recommendations, and the set-up of an international registry between 2014 and 2019. Based on the previous multicenter registry on mid-term procedural and clinical outcomes of percutaneous PVL closure with PLD [17], we sought to assess the long-term efficacy and safety of this specifically designed device for PVL closure and the impact of these procedures on clinical outcome and quality of life of the patients.

## 2. Materials and Methods

This study included PLD closure procedures performed in 21 hospitals in 10 EU and non-EU countries, as previously described [17]. Patients had PVL-associated hemolysis, recurrent blood transfusions or hemodynamically significant heart failure and were deemed at risk for surgical intervention. Patients had multiple co-morbidities, such as coronary artery disease, diabetes mellitus, hypertension, previous stroke, chronic renal failure, atrial fibrillation, rheumatic valve disease and previous infective endocarditis. Demographic data and medical history are summarized in Table 1.

Anonymized data were acquired from medical and electronic records regarding patient medical history, demographics, vital signs, clinical laboratory tests, 12-lead electrocardiography (ECG) and two- and three-dimensional (2D/3D) transthoracic/transesophageal echocardiography (TTE/TEE). Signed, informed consent was obtained from all patients prior to the procedure. The study plan was approved by an independent ethics committee, the International Medical and Dental Ethics Commission (IMDEC, Freiburg im Breisgau, Germany).

Comprehensive baseline 2D/3D TEE color Doppler was used to detect the precise location and size of PVLs and their relationship with the surrounding structures. Complementary imaging modalities like multidetector computed cardiac tomography (MDCT) and cardiac magnetic resonance (CMR) were also used for better morphologic and functional assessment [18]. In a restricted number of complex Mi and Ao PVLs, the anatomy was assessed before the procedure by 3D printing MDCT, and a silicone-printed 3D heart model was created to better understand leakage anatomy and its relationship with the surrounding structures and for planning ex vivo transcatheter implantation of the most appropriate occluding device (3D patient-specific simulation-guided treatment strategy) [19].

2D/3D TTE/TEE color Doppler flow mapping, including real-time 3D full volume acquisition and occasionally 3D-TrueVue (Philips Medical Systems, Amsterdam, The Netherlands) images, was used during the procedure. The degree of PVL regurgitation was evaluated by color Doppler TTE/TEE according to American Society of Echocardiography guidelines [20]. Procedures were performed under general anesthesia or conscious sedation due to the need for intraprocedural TEE guidance. In a subset of patients with Mi PVLs, the transapical hybrid approach was applied using EchoNavigator echocardiographic-fluoroscopic fusion imaging software (second release software, Philips Medical Systems, Best, The Netherlands). This imaging technique was particularly useful in patients with radiolucent biological mitral valves and for facilitating transcatheter treatment in the case of multiple PVLs [21,22].

The efficacy and safety of implanted devices were assessed by vital signs, laboratory tests, electrocardiogram (ECG) and TTE at 1 day (no later than 36 h post procedure), 30 days, 180 days and 2 years after the procedure. Patients were treated according to the Instructions for Use (IFU) of the device and clinical routine. Patients were screened to determine eligibility for the registry based on the following criteria: age > 18 years, history of surgical AVR or MVR complicated by symptomatic PVL (HF and/or HA requiring recurrent blood transfusions) with indication for a transcatheter procedure, high surgical risk after consultation with a surgeon or alternative to surgery with less operational time and recovery period. Exclusion criteria were as follows: active infective endocarditis or unexplained elevation of inflammatory markers, anemia related to factors other than hemolysis (bleeding, cancer, chronic inflammation), multiple PVLs with at least one deemed unsuitable for a transcatheter procedure (anatomy, size, location) or indications for surgical repair (prosthetic valve instability, structural deterioration of the prosthetic valve or need of coronary artery bypass grafting). The primary efficacy endpoint was successful implantation of the device with effective closure of the PVL (defined as reduction in paravalvular regurgitation of ≥1 grade, as assessed by echocardiography post implantation) and/or reduction in the number of hemolysis-related transfusions. The primary safety endpoint was the absence of serious adverse events (SAEs) at 180 days after the procedure. Additional secondary safety endpoints were minor complications that were deemed relevant at the discretion of each investigator. Further efficacy assessments were variation in the data from baseline to 6 months after the procedure in parameters such as, but not limited to, aortic blood pressure, pulse rate, left ventricular ejection fraction (LVEF), lactate dehydrogenase (LDH), blood count, regurgitation volume, regurgitation fraction, N-terminal pro-brain natriuretic peptide (NT-pro BNP) value and ECG.

Procedures were performed at sites having appropriate catheterization laboratory support and adequately trained imaging personnel by physicians with wide experience in interventional treatment of structural heart disease interventional cardiology and structural heart disease treatment, including trans-septal puncture and anterograde, retrograde or trans-apical approaches.

The statistical analysis comprised the safety and efficacy data of 137 SAF patients and efficacy data of 112 FAS patients. The FAS population was defined as patients with baseline and 180-day assessment and later. Follow-up data at 2 years was available in 80 FAS patients. Categorical variables are presented as numbers and percentages. Where appropriate, mean differences were compared using Student’s *t*-test for normally distributed variables. The Signed Wilcoxon test was used for non-normally distributed variables. Categorical variables were compared by Fisher’s exact test for 2 × 2 tables or by the sign test. For tables with more than 2 rows or columns, Pearson’s χ^2^ test was used. Two-sided *p*-values < 0.05 were considered statistically significant in all analyses. All calculations were carried out with SAS Version 9.4 (SAS Institute Inc., Cary, NC, USA).

## 3. Results

Procedural characteristics of patients with Mi and Ao PVLs are summarized in Table 2. 

This report covers PLD implantations in the period from 27 December 2014 until 23 January 2019; the cut-off for this analysis was 9 August 2021. The most common medical histories overall were atrial fibrillation (45 patients, 62.5%) and hypertension (38 patients, 52.8%). The percentage of patients with atrial fibrillation was greater in patients with mitral PVLs than in patients with aortic PVLs (81.8% vs. 32.1%, respectively). At the time of this analysis, 144 patients were enrolled at 21 sites in 10 countries: Poland (50 patients), Italy (34 patients), Lithuania (24 patients), France (16 patients), United Kingdom (7 patients), Cyprus (4 patients), Kazakhstan (4 patients), Hong Kong (2 patients), Hungary (2 patients), and Russia (1 patient).

Seven patients were excluded from both the SAF and FAS due to missing AE documentation and/or stratum issue. A further 25 patients were excluded from the FAS, which therefore comprised 112 patients with distinct stratum as well as baseline and Day 180 or later (2 years to 4 years) assessments for at least one of the parameters: NYHA, blood transfusion and/or regurgitation grade. Seventy-nine (57.7%) patients included in the SAF were male, while 58 patients (42.3%) were female. Mean age was 66.7 years (range 26 years to 85 years). The FAS comprised 112 patients with available stratum (Ao/Mi) as well as baseline, 180 day and later assessments. Ninety-three patients in the SAF had Mi PVLs, and 44 patients had Ao PVLs.

Procedural details of the FAS population have been reported elsewhere [17]. In summary, procedural success, defined as stable device implantation without moderate/severe residual leak and procedural complications, was achieved in 42/46 (91.3%) patients with Mi PVLs and in 18/20 (90.0%) patients with Ao PVLs.

The mean procedure time for Mi PVLs was 94.8 min for 1 leak (*n* = 45), 125.0 min (*n* = 20) for 2 leaks, and 170.0 min (*n* = 5) for 3 leaks. For Ao PVLs, the mean procedure time was similar for 1 leak (97.6 min, *n* = 29) and 2 leaks (90.0 min, *n* = 8) (Appendix A). Mean fluoroscopy time for Mi PVLs was 26.6 (*n* = 36), 29.7 (*n* = 18) and 36.0 min (*n* = 4) for 1, 2 and 3 leaks, respectively. For patients with Ao PVLs, mean fluoroscopy time was 19.5 min (*n* = 25) for 1 leak and 16.7 min (*n* = 7) for 2 leaks (Appendix A). Most of the leaks were closed on the first attempt (mitral: 85.4%; aortic: 82.4%). A second attempt was needed in 9.0% of Mi and 17.6% of Ao PVLs, while three attempts were needed in 2 (2.2%) patients with Mi PVLs.

Vital signs and laboratory values are summarized in Table 3.

No clinically significant changes were observed between baseline and follow-up at the primary timepoint of Day 180 for most of the variables tested. There was a small but statistically significant mean (SD) increase from baseline at Day 180 in diastolic blood pressure (+3.5 (0.3) mmHg) and a small but statistically significant increase from base-line in hemoglobin (+0.4 (1.3) mmol/L).

Mean (SD) NT pro BNP overall was 1611.2 (2680.2) pg/mL at baseline and decreased by 284.4 (877.5) pg/mL at Day 180. There was a marked decrease in NT pro BNP in patients with Ao PVLs, with a mean (SD) baseline value of 2094.8 (3508.6) pg/mL and reduction at Day 180 of 1036.7 (1919.8) pg/mL. Otherwise, variability was high, and changes from baseline were not observed consistently. NYHA functional class improved at 6 months and at 2 years compared with baseline (Figure 1).

At baseline, 80.0% of patients in the FAS had significant or severe heart failure (NYHA class III or IV). At 180 days after PLD implantation, only 10.9% of patients were NYHA class III or IV. The change from baseline was statistically significant (*p* < 0.0001). There were 78 patients for whom NYHA classification was available at 2 years. Of these, (85.9%) patients were in NYHA class I or II.

Qualitative changes in NYHA class from baseline to 180 days and to 2 years showed that the majority of patients (86.4% at 180 days and 80.8% at 2 years) had a clinical improvement (Appendix A), and this was observed in both Mi PVLs (84.9% and 82.4%) and Ao PVLs (89.2% and 77.8%) patients. The need for hemolysis-related blood transfusions was generally lower at 180 days compared with baseline and was similarly reduced during long-term follow-up (Figure 2A). At baseline, 26.4% of patients in the FAS received a hemolysis-related blood transfusion, while only 3.6% of patients needed blood transfusion at 180 days and 2.6% at 2 years after PLD implantation (22.0% and 16.7 decrease from baseline, respectively) (Figure 2B).

The percentage of patients with a decreased need for hemolysis-related blood transfusions from baseline was 29.2% in patients with Mi PVLs and 8.1% in patients with Ao PVLs. There were 78 patients with hemolysis-related blood transfusion data at 2 years. Of these, 76 (97.4%) patients did not require blood transfusion.

The paravalvular regurgitation grade was assessed using a 2D/3D TTE/TEE color Doppler before PLD implantation and at follow-up, and was deemed ‘no’, ‘small’, ‘moderate’ or ‘severe’. A comparison of baseline and follow-up assessment at 180 days and 2 years is shown in Figure 3.

At baseline, 95.5% of patients in the FAS had severe regurgitation, while 86.4% of patients had no or small regurgitation at 180 days. There were 77 patients with PVL regurgitation grading at 2 years. Of them, 71 (92.2%) patients had no or small residual leak.

Residual leak assessment in percentage from implantation to 6 months and 2 years is shown in Appendix A.

Clinical success was defined as patients with NYHA class I or II, and NYHA class at baseline > NYHA class at 6 months, or with blood transfusions at baseline and no blood transfusions at 6 months, and who did not experience a serious device- or procedure-related complication. Clinical success was achieved in 96/110 patients (87.3%; 95% confidence interval (CI): 79.6%, 92.4%) overall (Table 4, Figure 4).

Clinical success was achieved in similar proportions of patients with Mi and Ao PVL repairs: 63/73 patients (86.3%; 95% CI: 76.4%, 92.6%) with Mi PVLs and 33/37 patients (89.2%; 95% CI: 74.7%, 96.3%) with Ao PVLs. At Year 2, clinical success was achieved in 61/79 patients (77.2%; 95% CI: 66.8%, 85.2%).

A total of 50 adverse events (AEs) were reported in 37 patients (27.0%) in the SAF population, and 35 serious adverse events (SAEs) were reported in 27 (19.7%) patients. A total of six6SAEs were considered possibly or probably related to the device/procedure and included device embolization in three patients, residual leak in two patients and a vascular complication (pseudoaneurysm at femoral artery puncture site) in one patient. Among the three patients (2.2%) with device embolization, attempts to recapture the embolized device with the snare failed: two patients underwent redo surgery immediately after the PLD migrated in the left ventricle and in the left atrium, respectively, while the third patient underwent redo surgery three weeks after the procedure due to late partial detachment of one of the three devices implanted. Of the two patients (1.5%) with residual leaks, one patient underwent a repeated successful closure procedure, and the second one had residual paravalvular leak through the device, which caused severe anemia requiring four units of blood transfusion on a weekly basis.

Among thee 35 serious adverse events (SAEs), eight SAEs occurred in between implantation visits to follow-up 1, 10 were between follow-up 1 to follow-up 2, four SAEs were between follow-up 3 to follow-up 4, and 13 SAEs were between follow-up 4 to follow up 2 years (Figure 5).

There were no SAEs attributed to the Occlutech delivery set and Occlutech pistol pusher. During follow-up, 11 of 137 (8.8%) patients died. No death was considered to be device-related.

## 4. Discussion 

Transcatheter PVL closure in symptomatic patients following surgical valve replacement is generally recommended in those with multiple comorbidities, who are often ineligible or unsuitable for redo surgery. However, the paucity of closure devices specifically designed for this purpose has hampered PVL interventional treatment. Self-expanding devices, mostly Amplatzer vascular plugs (AVP) II/III/IV, duct occluders (ADO) I/II, atrial septal occluders (ASO) and muscular ventricular septal defect occluders (MVSDO), are used off-label, depending on patient’s anatomy. Importantly, the Occlutech PLD, which is available in Europe (CE marked in 2014), is the first device to be designed specifically for PVL closure. The PLD is a self-expanding, double disc device made from Nitinol braided wires. The two discs are specifically designed not to overlap with the valve area and are linked together by a waist equivalent to the PVL size. The study population comprised patients requiring percutaneous PVL closure. The mean age was 66.7 years, and 57.7% were male. The mean LVEF was 49.5%, which is slightly lower than would be expected in the normal population (50 to 70%). A reduced LVEF can be explained by the fact that most patients had cardiac failure NYHA class III or IV and severe valvular regurgitation at baseline. Overall, PVLs have irregular geometries and a complex surrounding anatomical environment and can vary significantly in size and shape (crescent, oval, round or slit). Multiple leaks are not uncommon.

PVL tracks can be parallel or perpendicular, but mostly serpiginous and tortuous. Of the patients with Mi PVLs, 28.4% had two leaks and 6.8% had three leaks, while 21.1% of patients with Ao PVLs had two leaks. Considering the complexity of PVL geometries, the availability of a range of PLDs with different sizes, shapes and waists is extremely important for achieving procedural success. Indeed, PLD is available in four different shapes and 19 different sizes in order to conform to the variety of leak shapes and sizes (Appendix A). In this registry, approximately 49% of Mi PVLs and 67% of Ao PVLs were crescent-shaped, a quite difficult shape to be closed with round atrial or ventricular septal defect devices, which tend to interfere with the valve leaflets. In line with the frequency of the crescent-shaped leaks observed in this study, rectangular-shaped PLD devices were predominantly used (63% of patients with Mi PVLs and 77% of patients with Ao PVLs) (Table 2).

A further advantage of the rectangular-shaped PLD is the option to close large defects (Figure 6) or even multiple leaks located in close proximity with one single device.

Indeed, in 87.5% of the patients with two Ao PVLs, one single device was used for leak closure. Similarly, in 33.3% of the patients with two Mi PVLs and in 20.0% of patients with three Mi PVLs, fewer devices than the number of leaks were used. This is a clear advantage compared to using multiple smaller devices in terms of stability, procedure-time and costs and confirms the findings previously reported by Goktekin et al. [23], who reported that the rectangular shape was associated with a higher success rate. The primary performance endpoint of this study is the proper closure of the PVL, defined as a reduction in paravalvular regurgitation by ≥1 grade at 180 days post implantation and/or reduction in the number of hemolysis related transfusions 6 months after the procedure.

Our study met this endpoint in 94.5% of the patients at 180 days. In previous studies with off-label devices, technical success rates varied between 62% [24] and 87% [6]. In 20.9% of our patients, paravalvular regurgitation was no longer detectable with echocardiography assessment at 180-day follow-up.

Several studies have shown a direct correlation between the grade of regurgitation and repeat surgical interventions and survival rates [25,26], reaffirming the concept that residual PVL regurgitation may represent the key determinant of outcome.

The excellent technical success was mirrored by a significant improvement in terms of NYHA class following PLD implantation. At baseline, 80.0% of the patient cohort had heart failure (NYHA class III or IV), a rate that decreased to 10.9% at 180 days after PVL closure. This effect was maintained at long-term follow-up. Indeed, at 2 years post implantation, 85.9% of the patients were in NYHA class I or II (Figure 1). In total, 86.4% of the patients at 180 days and 80.7% at 2 years had an improvement in NYHA class compared to baseline (Appendix A).

Similarly, mean NT-pro BNP was lower (284.4 pg/mL decease) at 180 days than at baseline (1.612 pg/mL), indicating an improvement in cardiac function. However, the decrease from baseline was not statistically significant, probably due to the high variability of the data.

The need for hemolysis-related blood transfusions was reported in 26.4% of the patients at baseline. Overall, a reduction in the need for hemolysis-related blood transfusions at 180 days compared to baseline was observed in 22.0% of patients (29.2% of patients with Mi PVLs and 8.1% of patients with Ao PVLs). At 2 years, a decrease in hemolysis-related blood transfusions compared to baseline was reported in 16.7% of patients (Figure 2).

There are several potential AEs that might occur during PVL closure, including (1) prosthetic valve impingement, which can be expected to be more frequent due to the proximity of the device to the valve leaflets; (2) higher instability and device embolization when multiple devices have to be deployed in large PVLs; (3) incomplete PVL closure, which may lead to hemolysis and subsequent anemia. Indeed, procedural complications such as device malpositioning and even embolization are observed frequently [9,11]. This might occur if inappropriate device sizes/designs are chosen or the device is dislocated during the procedure [12,14]. Generally, in such cases, withdrawal of the device into the delivery sheath can be performed. However, if the interventional cardiologist fails to snare the device, emergency surgery might be necessary.

In our study, a total of 50 AEs were reported in 37 (27.0%) patients, including three device embolizations, two residual leaks and one procedural vascular complication. Device migration is not uncommon in PVL closure procedures, due to weakened tissue around the valve, multiple-device deployments and complicated access to PVLs combined with limited visualization options. Ruiz et al. reported an embolization rate with an off-label device of 4.7% [13].

The vast majority of patients in whom a PLD device was implanted for PVL closure significantly benefited from the procedure, as indicated by NYHA class improvement and a lower need of hemolysis-related blood transfusions.

Certainly, most PVL patients have one or more comorbidities (coronary artery disease, peripheral vascular disease, hypertension, atrial fibrillation, renal insufficiency, chronic obstructive pulmonary disease, prior stroke, coronary artery bypass grafts or permanent pacemaker) [25,26]. As expected and due to the relatively high mortality risk of this patient population, 12 (8.8%) deaths were reported in this registry. This rate is comparable to those reported in the literature [9,11]. Of note, none of the deaths in this registry was considered related to the device.

Surgical repair or valve replacement is still regarded as the treatment of choice for patients with significant PVL. Nevertheless, redo surgery should be recommended if prosthetic valve regurgitation is related to recurrent endocarditis or causes hemolysis, requiring repeated blood transfusions or leading to severe symptoms.

Recently, transcatheter PVL closure has emerged as a reliable alternative therapy that ensures at least similar clinical outcomes in patients for whom redo surgery is not suitable or is associated with prohibitive risk. Catheter-based techniques have been endorsed by ACC/AHA and ESC guidelines for symptomatic PVL management [27,28]. Therefore, transcatheter closure may be considered a step-wise approach for high-risk patients that offers a less invasive approach before performing a surgical intervention. Indeed, it should be also emphasized that a transcatheter attempt at PVL closure does not preclude a subsequent surgical intervention.

PVL treatment requires careful preprocedural imaging, planning and patient selection, and it should be performed in high-volume referral centers. Undoubtedly, transcatheter PVL closure remains a technically demanding procedure that might be facilitated by a close collaboration between imaging experts and interventional cardiologists (“team approach”).

Furthermore, the use of multi-modality imaging such as fusion of real-time 3D TEE and cardiac fluoroscopy imaging has been shown to improve the safety and efficacy of percutaneous Mi and Ao PVL closure, facilitating catheter and guidewire access through the leaks by target markers [29] (Figure 7A,B; Appendix A), reducing procedural time and radiation exposure.

New techniques such as 3D printing and modelling are gaining relevance in cardiovascular medicine, offering possibilities for pre-procedural evaluation and simulation of the procedures (Appendix A). Further technical advances, an increasing interventional experience and future developments in imaging technology will help to consolidate this alternative option for high-risk patients.

Our registry has certain limitations that must be considered. First, the inclusion of patients in this study was not standardized prospectively. Therefore, there is a theoretical bias associated with such an investigation. Second, based on the retrospective character and the large number of patients, there was no assessment of the images by a central core laboratory, nor was an audit of the records performed. However, all data were continuously collected on an intention-to-treat basis according to the quality assurance program of the participating hospitals. Similarly, adverse events, problems during implantation and follow-up data were monitored. AEs during follow-up were reported by the submitting centers based on the local follow-up program of the referring doctors. Third, different hospitals and investigators with different skills and techniques participating in this study increased the complexity of comparison of the implantation methods used. Fourth, the sample size, especially at later follow-up, and the follow-up period may be judged only as moderate. However, the age range as well as the different sizes and anatomies of the PVLs presented seem representative of a normal patient cohort for interventional closure procedure.

## 5. Conclusions

The data collected in this study demonstrate that percutaneous closure of PVLs with the Occlutech PLD is a safe and viable therapeutic alternative to surgical PVL repair, with a high technical success rate in both Mi and Ao PVLs and a low rate of recurrent or residual leaks. Patients implanted with the Occlutech PLD showed significant acute and long-lasting improvements in clinical parameters, including NYHA class, and a reduced dependency on hemolysis-related blood transfusions.

## Figures and Tables

**Figure 1 jcm-11-01978-f001:**
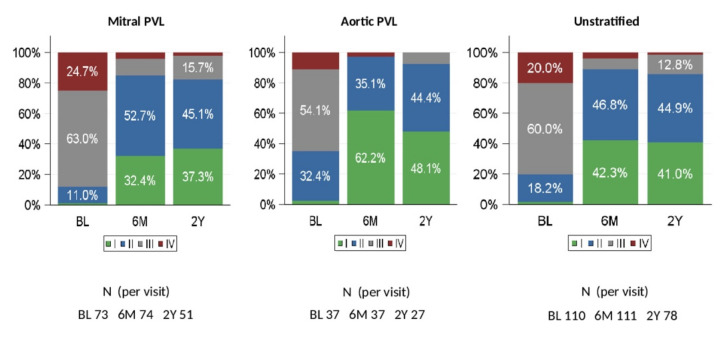
NYHA Class (relative frequencies) at baseline, 6 months, and 2 years (FAS). BL: baseline, 6M: 6 months; 2Y: 2 years. NYHA = New York Heart Association. PVL = paravalvular leak. FAS= full analysis set.

**Figure 2 jcm-11-01978-f002:**
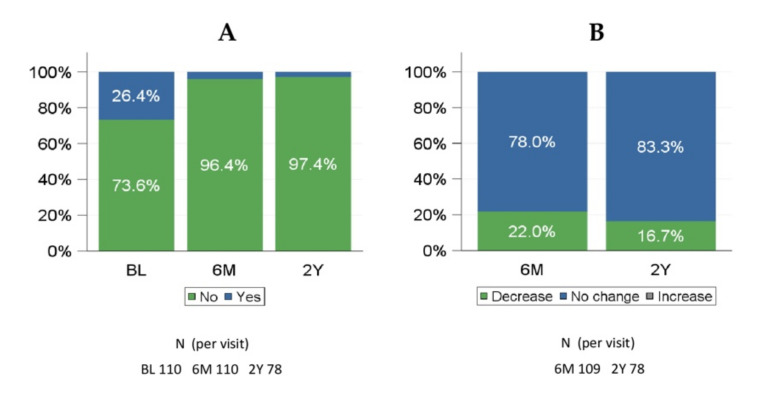
Hemolysis-related blood transfusions at baseline, 6 months, and 2 years (FAS). (**A**) any hemolysis-related blood transfusion given; (**B**) qualitative change from baseline in hemolysis-related transfusions. 6M: 6 months; 2Y: 2 years. FAS = full analysis set.

**Figure 3 jcm-11-01978-f003:**
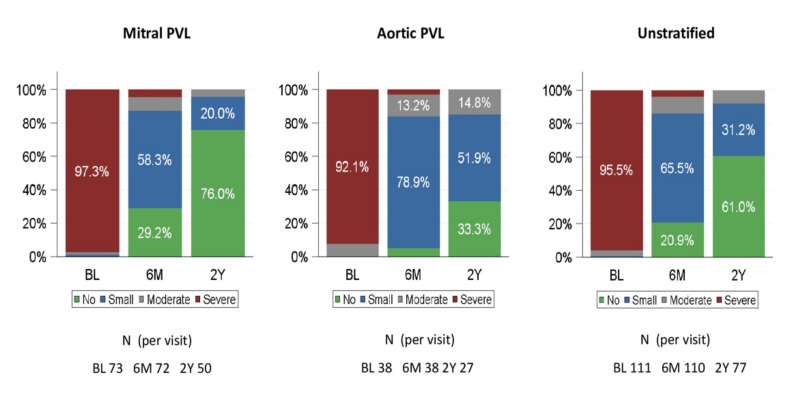
Paravalvular regurgitation grade at baseline, 6 months and 2 years (FAS). BL: baseline, 6M: 6 months; 2Y: 2 years. PVL = paravalvular leak. FAS = full analysis set.

**Figure 4 jcm-11-01978-f004:**
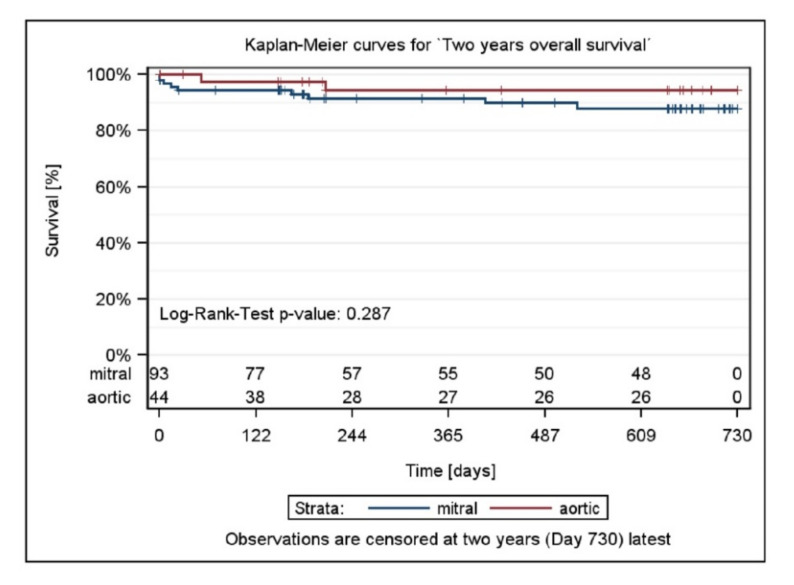
Two-year survival by stratum and total and *p*-value testing for homogeneity.

**Figure 5 jcm-11-01978-f005:**
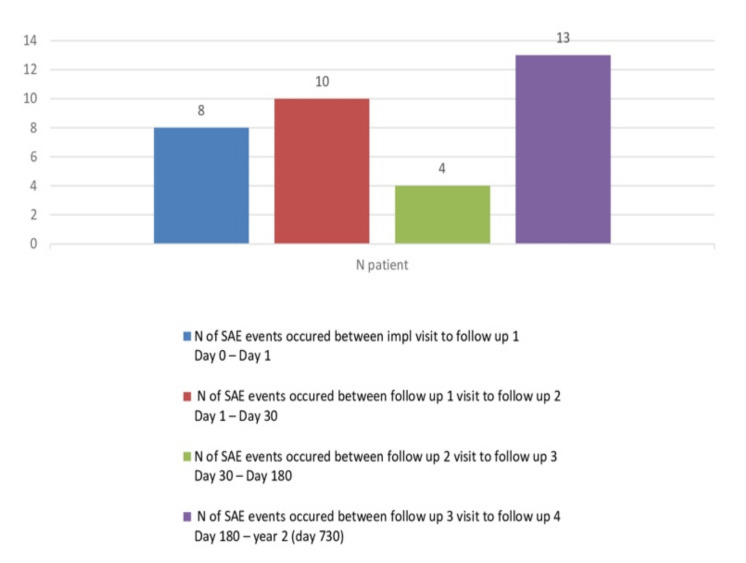
Number of SAE events occurred between visits (FAS).

**Figure 6 jcm-11-01978-f006:**
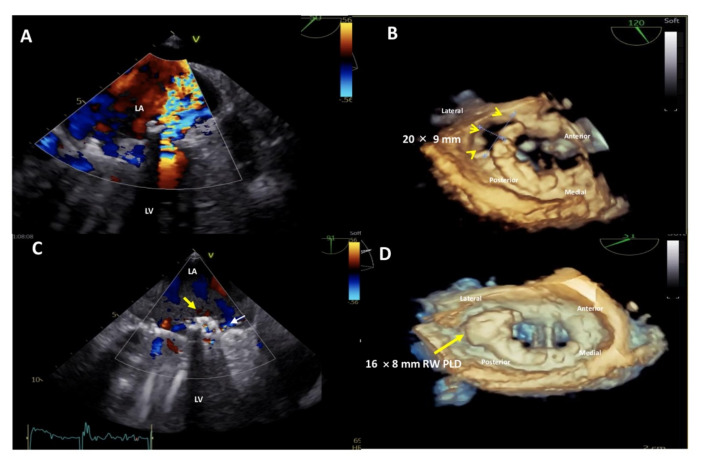
Upper panel: baseline 2D Transesophageal Echocardiogram (TEE) color Doppler showing a severe paravalvular leak (PVL) regurgitant jet into the left atrium (**A**) and 3D TEE showing a huge, 20 × 9 mm in diameter, crescent-shaped, antero-laterally located (7–10 o’clock) mitral PVL after bileaflet mechanical valve replacement (**B**). Lower panel: post-procedure 2D TEE color Doppler showing the correct position of the device (**C**, yellow arrow) with a trace-mild residual regurgitant jet (**C**, small white arrow) and 3D TEE confirming the stable position of the 16 × 8 mm-rectangular waist PLD (yellow arrow), without impingement on the mechanical prosthetic mitral valve (**D**). LA, left atrium; LV, left ventricle; PLD, Occlutech Paravalvular Leak Device; RW, rectangular waist.

**Figure 7 jcm-11-01978-f007:**
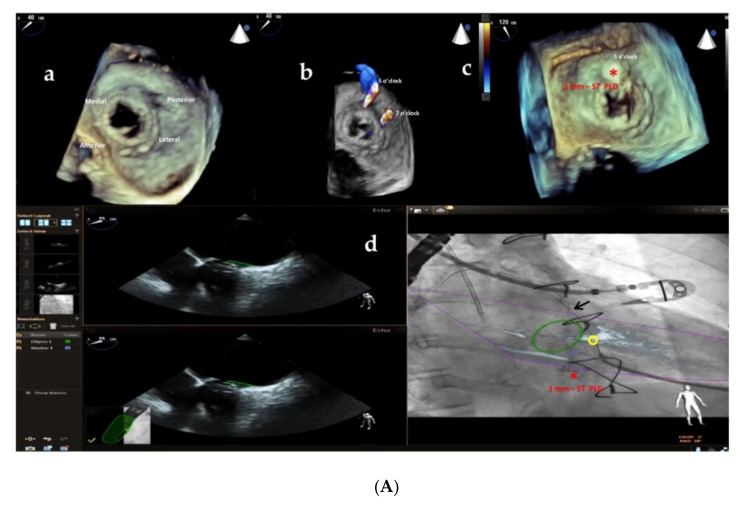
(**A**) Baseline 3D TEE color Doppler showing two round-shaped mitral PVLs with significant regurgitant jets located at 5 o’clock (3.2 mm) and 7 o’clock (3.6 mm) (**a**,**b**) in a female patient with a mitral bioprosthetic valve replacement; (**c**): successful implantation of a 3 mm-square twist PLD (red asterisk) at 5 o’clock; (**d**): fusion of real-time 2D TEE (Phillips Epiq7) and cardiac fluoroscopy imaging was obtained using EchoNavigator®-system (Philips Healthcare, Best, The Netherlands) with the fused image maintained demonstrating the location of the postero-laterally located (7 o’clock) mitral PVL (yellow circle) that was marked to aid the pathway of the guidewire (black arrow) crossing the leak, particularly in the case of a radiolucent bioprosthetic valve like in our patient. “*” refers to 3 mm-square twist PLD. (**B**) Intraprocedural navigation using fusion of 3D TEE and fluoroscopy with the fused image maintained helped the interventionalist to close successfully the second leak with a 5 mm-square twist PLD (black asterisk), improving the safety and the efficacy of this technically challenging procedure. PLD, Occlutech Paravalvular Leak Device ST, square twist, “*****” refers to 5 mm-square twist PLD.

**Table 1 jcm-11-01978-t001:** Demographic characteristics, gender and medical history of study patients in the safety analysis set (SAF).

	Mitral PVL (*n* = 93)	Aortic PVL (*n* = 44)	Total (*n* = 137)
	Number (%) of patients		
Gender Male	42 (45.2%)	37 (81.1%)	79 (57.7%)
Female	51 (54.8%)	7 (15.9%)	58 (42.3%)
Age (years) Mean (SD) Min, max	67.4 (8.5) 43, 85	65.2 (13.6) 26, 84	66.7 (10.4) 26, 85
Medical history	*n* = 44	*n* = 28	*n* = 72
Coronary artery disease	10 (22.7%)	7 (25.0%)	17 (23.6%)
Diabetes mellitus	7 (15.9%)	7 (25.0%)	14 (19.4%)
Hypertension	21 (47.7%)	17 (60.7%)	38 (52.8%)
Atrial fibrillation	36 (81.8%)	9 (32.1%)	45 (62.5%)
Chronic Obstructive Pulmonary Disease	3 (6.8%)	1 (3.6%)	4 (5.6%)
Prior stroke	4 (9.1%)	0 (0.0%)	4 (5.6%)
Chronic renal failure	10 (22.7%)	1 (3.6%)	11 (15.3%)

PVL = paravalvular leaks. SAF = safety analysis fraction.

**Table 2 jcm-11-01978-t002:** Procedural characteristics in the full analysis set (FAS).

	Mitral PVL (*n* = 74)	Aortic PVL (*n* = 38)
	Number (%) of patients or leaks	
Approach	Missing data, 9	Missing data, 2
Antegrade trans septal	21 (32.3%)	0 (0.0%)
Retrograde trans aortic	2 (3.1%)	36 (100%)
Retrograde trans apical	42 (64.6%)	0 (0.0%)
Number of devices used per leak	Missing data, 11	Missing data, 7
1	84 (89.4%)	37 (94.9%)
2	9 (9.6%)	2 (5.1%)
3	1 (1.1%)	-
Number of devices used for all leaks per patient	Missing data, 1	Missing data, 0
1	50 (68.5%)	35 (92.1%)
2	16 (21.9%)	3 (7.9%)
3	5 (6.8%)	-
4	2 (2.7%)	-
Occluder shape per occlude	Missing data, 19	Missing data, 2
Square Waist	10 (11.6%)	2 (5.1%)
Square Twist	9 (10.5%)	1 (2.6%)
Rectangular Waist	54 (62.8%)	30 (76.9%)
Rectangular Twist	13 (15.1%)	6 (15.4%)

PVL = paravalvular leaks. FAS = full analysis set.

**Table 3 jcm-11-01978-t003:** Baseline and changes from baseline to Day 180 in vital signs and laboratory evaluations (FAS).

	Mitral PVL (*n* = 74)		Aortic PVL (*n* = 38)		Total (*n* = 112)	
	Baseline	Change from Baseline to Day 180	Baseline	Change from Baseline to Day 180	Baseline	Change from Baseline to Day 180
Pulse rate (bpm) Mean (SD) Min, max	*n* = 66 77.8 (9.1) 60, 100	*n* = 59 1.4 (13.5) −20, 49	*n* = 37 72.6 (9.1) 52, 86	*n* = 32 −1.5 (11.7) −25, 29	*n* = 103 76.0 (9.4) 52, 100	*n* = 91 0.4 (12.9) −25, 49
Systolic blood pressure [mmHg] Mean (SD) Min, max	*n* = 66 125.5 (16.8) 90, 160	*n* = 59 1.4 (16.3) −45, 40	*n* = 37 130.1 (14.3) 100, 160	*n* = 31 0.5 (21.4) −40, 60	*n* = 103 127.2 (16.0) 90, 160	*n* = 90 1.1 (18.1) −45, 60
Diastolic blood pressure [mmHg] Mean (SD) Min, max	*n* = 66 72.0 (10.0) 40, 90	*n* = 59 3.6 (10.3)^t,WIL^ −20, 30	*n* = 37 68.6 (9.5) 40, 90	*n* = 38 3.4 (10.5) −20, 20	*n* = 103 70.8 (9.9) 40, 90	*n* = 90 3.5 (10.3)^t,WIL^ −20, 30
LVEF [%] Mean (SD) Min, max	*n* = 70 49.2 (9.2) 29, 70	*n* = 66 1.1 (7.1) −25, 15	*n* = 37 50.1 (11.0) 20, 70	*n* = 37 0.8 (8.6) −18, 18	*n* = 107 49.5 (9.9) 20, 70	*n* = 103 1.0 (7.6) −25, 18
LDH [U/L] Mean (SD) Min, max	*n* = 34 716.8 (702.4) 100, 3057	*n* = 25 −111.9 (625.6) −1990, 1230	*n* = 13 319.8 (174.6) 100, 766	*n* = 6 16.3 (300.2) −490, 435	*n* = 47 607.0 (627.8) 100, 3057	*n* = 31 −87.1 (575.1) −1990, 1230
Erythrocytes [Mio/μL] Mean (SD) Min, max	*n* = 46 4.0 (0.8) 3, 6	*n* = 36 0.2 (0.9) −1, 2	*n* = 29 4.1 (0.6) 3, 5	*n* = 15 0.0 (0.8) −1, 1	*n* = 75 4.0 (0.7) 3, 6	*n* = 51 0.2 (0.9) −1, 2
Thrombocytes [Thsd/μL] Mean (SD) Min, max	*n* = 48 199.0 (68.3) 107, 409	*n* = 36 15.0 (62.5) −111, 164	*n* = 29 194.9 (51.0) 77, 306	*n* = 16 −2.4 (44.9) −100, 61	*n* = 77 197.4 (62.0) 77, 409	*n* = 52 9.6 (57.8) −111, 164
Leucocytes [/μL] Mean (SD) Min, max	*n* = 46 6605.2 (2124.4) 3400, 12630	*n* = 36 465.2 (2179.4) −3410, 8090	*n* = 29 6838.3 (1593.9) 4090, 10360	*n* = 15 −479.0 (1758.1) −3250, 2610	*n* = 75 6695.3 (1928.4) 3400, 12630	*n* = 51 187.5 (2092.6) −3410, 8090
Hemoglobin [mmol/L] Mean (SD) Min, max	*n* = 48 7.3 (1.4) 5, 10	*n* = 37 0.6 (1.3)^t,WIL^ −2, 3	*n* = 29 7.9 (1.2) 5, 11	*n* = 17 0.1 (1.1) −2, 1	*n* = 77 7.5 (1.4) 5, 11	*n* = 54 0.4 (1.3)^t,WIL^ −2, 3
NT-pro-BNP [pg/mL] Mean (SD) Min, max	*n* = 27 1467.9 (2446.7) 40, 11340	*n* = 22 −181.9 (661.2) −2230, 1132	*n* = 8 2094.8 (3508.6) 174, 9800	*n* = 3 −1036.7 (1919.8) −3238, 290	*n* = 35 1611.2 (2680.2) 40, 11340	*n* = 25 −284.4 (877.5) −3238, 1132

^t^ = *p* < 0.05 for one-sample *t*-test, ^WIL^ = *p* < 0.05 for Wilcoxon signed rank test. PVL = paravalvular leaks. FAS = full analysis set.

**Table 4 jcm-11-01978-t004:** Clinical success (FAS).

	Mitral PVL	Aortic PVL	Unstratified
Clinical Success at Day 180	*n* = 73	*n* = 37	*n* = 110
Yes (95% CI) ^1^	63 (86.3%) (76.4%, 92.6%)	33 (89.2%) (74.7%, 96.3%)	96 (87.3%) (79.6%, 92.4%)
Clinical Success at Year 2	*n* = 52	*n* = 27	*n* = 79
Yes (95% CI) ^1^	40 (76.9%) (63.7%, 86.4%)	21 (77.8%) (58.9%, 89.7%)	61 (77.2%) (66.8%, 85.2%)

^1^ Two-sided 95% Agresti Coull confidence interval. FAS = full analysis set

## Data Availability

The data presented in this study are contained within the article.

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
