# Peer review of "Safety, Efficacy and Long-Term Outcomes of Patients Treated with the Occlutech Paravalvular Leak Device for Significant Paravalvular Regurgitation"

_jcm, 2022, doi:10.3390/jcm11071978_

Round 1
Reviewer 1 Report
Onorato et al. evaluated the safety, efficacy as well as the medium to long-term outcomes of patients undergoing paravalvular leak (PVL) closure with the Occlutech device. The present analysis aims at reporting an extended follow-up of patients included in a retrospective, multicenter registry, whose acute and mid-term results have already been published in Eurointervention in 2020.
The object of the present study is indeed of clinical interest, since it adds relevant information regarding the outcomes of patients undergoing PVL closure procedures performed by experienced operators. Such procedures, which have received a lesser degree of attention in the medical literature compared to other structural heart interventions, often represent the only treatment option of patients for whom a repeat surgical procedure would portend a very high or prohibitive risk.
The manuscript is well written, the sample size quite large, the methodology appears solid and the conclusions supported by the results; moreover, the authors have nicely recognized the major strengths and limitations of the present work.
Comments:
- The completeness of clinical and echocardiographic follow-up appears not clearly depicted. Please report exact numbers of patients for whom clinical and echocardiographic follow-up data were available at 6 months and 2 years respectively.
- 3-year follow-up was available in only 7 patients and 4-year follow-up in only 1 patient. Since I do not believe that these small numbers are representative of the overall patient cohort, I would suggest to remove these data from the manuscript and censor the data at 2-year follow-up.
- Please show the Kaplan-Meier curves for mortality at 2 years.
- The figures reporting NYHA functional class, need for hemolysis-related blood transfusions as well as PVL regurgitation grade report only the relative frequencies. Please add the absolute patient numbers for whom the respective data were available at each time point (baseline, 6 months, 2 years).
Author Response
Milan, 26.03.2022
To
Journal of Clinical Medicine
Authorship Coverletter
We present this cover letter to explain, point by point, the details of the revisions to the manuscript (Article. Safety, Efficacy and Long-term Outcomes of Patients Treated with the Occlutech Paravalvular Leak Device for Significant Paravalvular Regurgitation) and our responses to the referees’ comments.
Review Report Form 1
Open Review
Onorato et al. evaluated the safety, efficacy as well as the medium to long-term outcomes of patients undergoing paravalvular leak (PVL) closure with the Occlutech device. The present analysis aims at reporting an extended follow-up of patients included in a retrospective, multicenter registry, whose acute and mid-term results have already been published in Eurointervention in 2020.
The object of the present study is indeed of clinical interest, since it adds relevant information regarding the outcomes of patients undergoing PVL closure procedures performed by experienced operators. Such procedures, which have received a lesser degree of attention in the medical literature compared to other structural heart interventions, often represent the only treatment option of patients for whom a repeat surgical procedure would portend a very high or prohibitive risk.
The manuscript is well written, the sample size quite large, the methodology appears solid and the conclusions supported by the results; moreover, the authors have nicely recognized the major strengths and limitations of the present work.
Comments:
- The completeness of clinical and echocardiographic follow-up appears not clearly depicted. Please report exact numbers of patients for whom clinical and echocardiographic follow-up data were available at 6 months and 2 years
√ Appropriate changes according to your suggestion have been made in the manuscript
- 3-year follow-up was available in only 7 patients and 4-year follow-up in only 1 patient. Since I do not believe that these small numbers are representative of the overall patient cohort, I would suggest to remove these data from the manuscript and censor the data at 2-year follow-up.
√ Appropriate changes according to your suggestion have been made in the manuscript
Please show the Kaplan-Meier curves for mortality at 2 years.
√ Appropriate changes according to your suggestion have been made in the manuscript, in particular a figure has been added to the manuscript (Figure 4)
- The figures reporting NYHA functional class, need for hemolysis-related blood transfusions as well as PVL regurgitation grade report only the relative frequencies. Please add the absolute patient numbers for whom the respective data were available at each time point (baseline, 6 months, 2 years).
√ Appropriate changes according to your suggestion have been made in the manuscript
Truly yours
Eustaquio Maria Onorato MD, FSCAI, FESC
Interventional Cardiology Dept, Centro Cardiologico Monzino,
IRCCS, Via C. Parea, 4 - 20138 Milan (Italy)
E-mail: eustaquio.onorato@gmail.com Phone: +39 348 693988
Reviewer 2 Report
In the current manuscript, Onorato and colleagues present safety, efficacy and long-term outcome data from patients treated with the Occlutec paravalvular leak device for paravalvular leakage
Especially the results section is very confusingly written, so I have several aspects that need further attention:
- What does Part II mean in the title? Each manuscript should stand on its own
- Please define procedural success in the abstract
- It is really unfortunate that a significant amount of data is missing, making the manuscript much more difficult to follow
- Why were the patients undergoing valve surgery? PVL is usually observed in patients with enodcarditis, please discuss this in the introduction and provide more baseline information about the patients.
- Patient recruitment started in 2014, so the longest possible follow-up would be 6-7 years. Why do you only have 6 patients with a follow-up of more than 3 years?
- Figure 4-7 do not contain any additional information as you are not describing a feasiblity study.
- Please change the colors of Figure 1-3 as they are too bright and not comfortable for colleagues with red green color vision deficiency
- When did the SAEs occur?
Author Response
Review Report Form 2
Open Review
In the current manuscript, Onorato and colleagues present safety, efficacy and long-term outcome data from patients treated with the Occlutec paravalvular leak device for paravalvular leakage
Especially the results section is very confusingly written, so I have several aspects that need further attention:
- What does Part II mean in the title? Each manuscript should stand on its own
√ Appropriate changes according to your suggestion have been made in the manuscript
- Please define procedural success in the abstract
√ Appropriate changes according to your suggestion have been made in the manuscript
- Why were the patients undergoing valve surgery? PVL is usually observed in patients with enodcarditis, please discuss this in the introduction and provide more baseline information about the patients.
√ Appropriate changes according to your suggestion have been made in the manuscript. Patients underwent valve surgery due to native valve degeneration, calcification or infective endocarditis.
- Patient recruitment started in 2014, so the longest possible follow-up would be 6-7 years. Why do you only have 6 patients with a follow-up of more than 3 years?
- It is really unfortunate that a significant amount of data is missing, making the manuscript much more difficult to follow
√ All patients were enrolled retrospectively (enrollment started in 2016), meaning that the available patient records/patient data were not designed for the study.
√ No monitoring was performed within the PLD Registry (CIP version 1.1 final, dated 19 Jan. 2016) which may have led to this amount of missing data.
- Figure 4-7 do not contain any additional information as you are not describing a feasiblity study.
- √ Appropriate changes according to your suggestion have been made in the manuscript
- Please change the colors of Figure 1-3 as they are too bright and not comfortable for colleagues with red green color vision deficiency
- √ Appropriate changes according to your suggestion have been made in the manuscript
- When did the SAEs occur?
- √ Appropriate changes according to your suggestion have been made in the manuscript. In particular a new table (Table 5) has been added in the manuscript at this purpose
Round 2
Reviewer 2 Report
Thank you for submitting your revised manuscript. : I congratulate the authors for their vastly improved manuscript. Concerns have been adressed.